# AI Ethics Education for Scientists

**Savannah Thais**
Data Science Institute
Columbia University
New York City, NY 10027
`st3565@columbia.edu`

## Abstract

Machine learning (ML) and artificial intelligence (AI) are becoming core components of scientific research across fields. While there are increasingly formal and informal domain-specific learning opportunities for students and early career scientists interested in AI/ML, AI ethics is often an overlooked part of these trainings. This is a concerning practice as a knowledge of the ethical considerations around AI/ML is an essential component of training effective and responsible scientists. This work presents a introductory AI Ethics curriculum tailored for scientists and describes implementations of the curriculum in various training scenarios.

## 1 Motivation

As computing power, the availability of software tools, and the capabilities of machine learning (ML) and artificial intelligence (AI) models have advanced, scientists across disciplines are increasingly incorporating AI/ML as integral components of their research programs. In fact, a recent Nature study found that nearly 10% of both life science and physical science papers published in 2023 mention AI or ML in their title or abstract [1]. Consequently, the demand for AI/ML educational materials and opportunities specifically tailored for scientists has also increased; the same Nature study found that over 60% of scientists using AI/ML in their research cite a lack of training resources as a barrier to developing or using AI/ML as much as they would like [1]. Several formal courses in AI/ML or data science are now offered as part of physics [2] and life science curricula [3, 4] and many summer schools and training workshops focused on ML/AI for science have been introduced in recent years.

While AI Ethics, Data Science Ethics, Data Science and Society, or similar courses are now typically a required component of formal data science and AI/ML programs [5, 6, 7], these topics are often neglected in AI/ML education materials for scientists. We argue that AI Ethics should also be an essential part of science-focused AI/ML training programs for four key reasons:

**1. Improved science.**   Many topics covered in AI Ethics curricula such as explainable AI, transparency and documentation, model evaluation, and data bias are critical tools for ensuring AI/ML research in science is reproducible, robust, and trustworthy. For example, the Reporting Standards for ML-based Science (REFORMS) checklist [8], which was recently proposed as a framework to help researchers avoid failures of validity, reproducibility, and generalizability in ML-based science, draws on many practices and methodologies developed in AI ethics scholarship.

**2. Job preparation.**   It is often difficult for scientists to identify direct societal implications of their AI/ML work within a specific scientific research project, and this frequently leads to a discounting of AI Ethics topics during scientific AI/ML training. However, many PhD scientists will eventually leave scientific research [9] and may seek to draw on their AI/ML skills to acquire jobs in industry. It is thus in the best interest of scientific trainees to develop the skills and knowledge necessary to critically evaluate a wide array of AI/ML methods and applications.

NeurIPS 2023 AI for Science Workshop.

**3. Societal considerations.** AI/ML is having a profound impact on nearly all facets of scoeity. Scientists are themselves citizens and community members who are impacted by the development and deployment of AI/ML systems. As technical researchers with proximity to and potential impact on this technology, scientists have duty to understand the societal contextualization and implications of AI/ML.

**4. Collaboration building** As scientists gain familiarity with AI Ethics topics and methods, it is possible to form a positive feedback loop where scientific researchers assist in developing state-of-the-art solutions to critical ethical concerns. Indeed, scientists are in some ways uniquely equipped to contribute to AI Ethics research due to the data- and theory-rich nature of many scientific AI/ML applications and scientists' experience with experiment design, uncertainty quantification, and model evaluation [10, 11].

## 2   Teaching to Scientists

When developing AI Ethics training materials for scientists, it is necessary to balance several different considerations: covering enough of core AI Ethics topics such that students receive an accurate characterization of the field, teaching students relevant and transferable AI ethics skills and methodologies that can be utilized in their current work, and cultivating student 'buy-in' such that the students appreciate the importance of the subject and retain information. We present a collection of best practices for designing effective and informative AI Ethics lessons for scientific audiences.

- **Focus on transferable skills:** While it is important to cover both the quantitative/technical and qualitative/social components of AI ethics, it is often useful to spend a majority of class time focused on transferable tools and methods such as explainability, data bias, quantitative fairness, and model evaluation. These topics are more closely related to 'traditional' science topics (and thus may feel more familiar to students) and teach skills that may be directly applicable to students' current research.

- **Use domain specific examples/case studies:** When possible, it is helpful to explicitly demonstrate how AI Ethics methods can directly or indirectly be utilized in scientific research. For example, when discussing explainable AI methods, one could introduce a paper applying a specific method in a scientific context (such as this paper applying relevance propagation to a particle identification model [12]) and ask students to analyze the paper's results from a scientific viewpoint. This practice can also allow students to see themselves as potential contributors to AI Ethics work.

- **Explore connections across fields:** A challenge in AI Ethics work is that different fields may use the same word to represent different concepts. For example, the word 'bias' is a charged term that can have many different meanings from a purely statistical description of shift in a distribution to a societal notion of preference or prejudice. It is helpful to call out these linguistic tensions explicitly so that students are able to describe issues precisely and develop fluency to understand AI/ML applications across domains.

- **Demonstrate importance of topic:** Current and future AI/ML systems have profound ethical and societal implications. Providing topical real-world examples of AI/ML impacts helps students understand the relationship that AI Ethics has to their day-to-day lives and communities, in addition to their scientific work, and fosters deeper engagement with the material.

- **Platform ethics scholarship:** It is important for instructors to emphasize that AI Ethics is an active field of scholarship that often requires deep technical and non-technical skills and is equally as important as other fields. Highlighting current AI Ethics research and researchers helps students understand the depth of the field and engage with the topics on a scholarly level.

- **Provide further resources:** In addition to highlighting current research, instructors should provide resources and suggestion for further engagement with the material. This can include sharing information about organizations working in the space, useful readings, and possible opportunities to get involved in research or advocacy efforts. This allows students to integrate the material and topics into their own practices as scholars and community members.

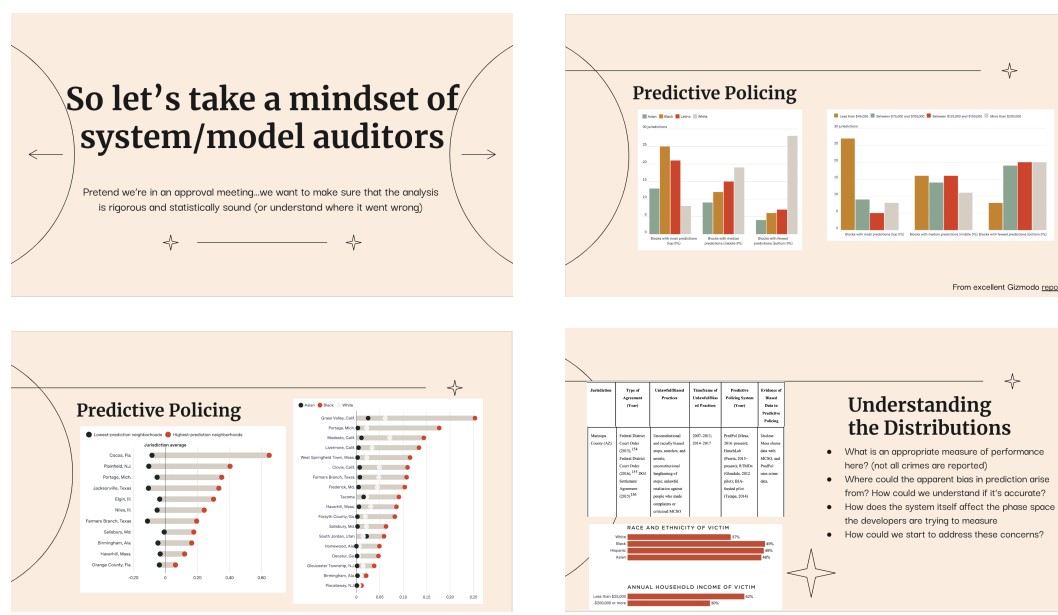

Figure 1: In-class discussion exercise prompting students to critically evaluate the output distributions of a predictive policing model

# 3 Curriculum Case Study

With these best practices in mind, we present a set of training materials specifically developed for teaching AI Ethics to scientists: `https://anonymous.4open.science/r/aiethics4science-6542/README.md`. The main component of this resource is a set 3 cohesive lectures and two guided, hands-on coding exercises, together providing six hours of dedicated content. This material is condensed into a one-hour, colloquim style presentation as well. Additionally, the repository includes the syllabus and initial lectures for a full-semester AI Ethics course currently being taught to Data Science Masters students. While data science is not a 'science' in the traditional sense, many of the same best practices discussed in Section 2 are implemented in this course and we believe it will be a useful resource for instructors teaching to scientific audiences.

The six hour course covers essential technical topics in AI Ethics topics including data bias, data collection and management, model evaluation and monitoring, explainability/interpretability, quantitative fairness, and privacy. It also discusses implications of using AI/ML in sensitive domains like determining resource access, predictive policing, risk assessments, and information curration, as well as societal considerations like regulation of AI/ML, participatory design, surveillance, AI/ML hype, and more. The full semester course dives deeper into these topics and also covers the ethical implications and considerations around current generative AI models, alternative philosophical frameworks for contextualizing AI/ML, technological determinism, and environmental impacts of AI/ML.

Both courses include hands on activities and discussion questions/prompts that encourage students to think about the topics within the context of their own work. For example, as shown in Figure 1, a discussion exercise asks students to evaluate performance distributions from a predictive policing tool just as they would examine and evaluate the results of a scientific experiment. Students are prompted to think through what factors in the data collection and model setup could impact these distributions and decide if they trust the claims of the original analysis.

## 3.1 Teaching Scenarios and Feedback

These materials have been used to provide AI Ethics instruction to scientific audiences in a variety of contexts including summer school lectures at the SLAC Summer Institute [13] and the Argonne Training Program on Extreme-Scale Computing [14], a series of classes provided to students in the SMART-HEP Network [15], and a full day course for the Large Synoptic Survey Telescope

Corporation Data Science Fellowship Program [16]. Although this is not a formal study evaluating the impact of these courses, we include a summary of common feedback received from students.

- Students report that the course strengthened their statistics skills, particularly around understanding data distributions and evaluating models, and that they feel better equipped to identify common technical pitfalls in model building.

- Students identify new methods and approaches to incorporate into their own research. This an especially common piece of feedback regarding the lesson on explainable AI methods, an area of AI Ethics research that has not been commonly applied to physical science models so far.

- Students are more aware of critical issues surrounding AI/ML technology and how it impacts their lives, their communities, and society more broadly. For many students this is their first time hearing some of the real-world examples presented in the course and they often express being shocked to learn about some of the societal impacts discussed.

- Student are inspired to learn more about various topics in AI Ethics. Individual students have asked for specific additional resources targeted to particular areas they are interested in, and some students have even gotten involved in reading groups or research collaborations focused on AI ethics.

- Students report feeling validated by the frank discussion of societal issues such as bias, oppression, and exploitation, topics that are rarely discussed in science classrooms. Students are able to integrate the material with their lived experiences, and often provide extremely insightful feedback that strengthens future iterations of the material.

## 4 Conclusions

As AI/ML becomes more entwined with society and scientific research, the importance of teaching AI Ethics to scientists grows. We must ensure that we are training scientists that have strong, generalizable data science and AI/ML skills that can be used across application areas and that they are prepared to be effective and responsible community members.

Incorporating AI Ethics into more AI/ML trainings for scientists not only improves the quality of scientific research, but also equips more people with the skills and insight necessary to help address some of society's most difficult questions around ethical technology. Scientists can play an important role in cultivating a more responsible and just technological future and this key educational opportunity should not be overlooked.

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
