# OpenReview forum: "AI Ethics Education for Scientists"
_NeurIPS.cc/2023/Workshop/AI4Science — NeurIPS2023-AI4Science Poster_

### Official Review · Reviewer_RZCB · 2023-10-23
**Incorporating AI Ethics into AI/ML trainings**

**Rating:** 6
**Confidence:** 4

**Review:**

This work explains why and how to incorporate AI Ethics into AI/ML trainings. It argues that AI Ethics worth essential part of AI/ML training for some important reasons. The paper has several pros and cons:

Pros:
1. Clear justification on the importance of involving AI Ethics into AI trainings.
2. Clear proposal on how to develop AI Ethics training materials for scientists.

Cons:
Personally I think this work lacks data to support the proposed solution.

---

### Official Review · Reviewer_MUAk · 2023-10-25
**Critical topic**

**Rating:** 10
**Confidence:** 5

**Review:**

The authors provide a very crucial topic for the current AI lanscape. As researchers, we are tasked with the resposibility to set a bechmark and be an example for other to follow. This AI Ethics curriculum proposed here is exactly what is needed. AI ethics courses will soon be a requirement like safety, HIPAA, and others. I commend the initiative from the scientists towards materialization of this important topic.

---

### Meta-Review · Area_Chair_EEv8 · 2023-10-27

**Recommendation:** Accept (Poster)
**Confidence:** 5

**Metareview:**

**Overview:**
The paper delves into the importance of AI ethics education within the realms of Machine Learning (ML) and Artificial Intelligence (AI) for scientists. The authors emphasize the critical need for integrating AI ethics into existing training modules, an area that's often overlooked. The paper also presents an introductory AI Ethics curriculum specifically designed for scientists, elaborating on its application in different training scenarios.

**Strengths:**

1. **Relevance and Timeliness:** The paper addresses an urgent need in today's AI-centric research environment. As AI/ML gets integrated across various scientific fields, understanding the ethical dimensions becomes paramount. The subject of AI ethics in the education of budding scientists is indeed timely.

2. **Curriculum Presentation:** The introduction of a tailored AI Ethics curriculum for scientists fills a significant gap in the current education landscape. This initiative can act as a benchmark for educational institutions and research organizations.

3. **Justification and Clarity:** The paper offers clear justifications for the inclusion of AI ethics in training programs. The proposal on developing training materials is laid out lucidly, making it easy for readers to understand the importance and the methodology.

**Areas of Concern:**

1. **Lack of Empirical Data:** While the paper presents an essential argument and solution, it seems to lack empirical evidence to substantiate its proposed curriculum. The inclusion of data, perhaps feedback from initial implementations or comparisons with other training modules, would have strengthened the paper's arguments.

2. **Implementation Scenarios:** While the paper mentions various training scenarios where the curriculum can be implemented, a deeper dive into these scenarios' specifics, challenges, and outcomes would have enriched the content.

**Overall Evaluation:**
Considering the feedback from both reviewers, the paper addresses a pressing issue in the current AI research environment and takes an important step in proposing a tailored curriculum. While the initiative is commendable and the proposal is clear, the paper would significantly benefit from the inclusion of supporting data and a more detailed exploration of the curriculum's real-world application.

**Recommendation:**
Given the relevance of the topic and the clear presentation of the curriculum, the paper is valuable. However, it would benefit from revisions that incorporate empirical evidence supporting the curriculum's efficacy. This would elevate the paper from being a foundational proposal to a comprehensive, evidence-backed recommendation for the scientific community.